# State of the Art of Non-Invasive Technologies for Bladder Monitoring: A Scoping Review

**DOI:** 10.3390/s23052758

**Published:** 2023-03-02

**Authors:** Abdelakram Hafid, Sabrina Difallah, Camille Alves, Saad Abdullah, Mia Folke, Maria Lindén, Annica Kristoffersson

**Affiliations:** 1School of Innovation, Design and Engineering, Mälardalen University, P.O. Box 883, 721 23 Västerås, Sweden; 2Textile Materials Technology, Department of Textile Technology, Faculty of Textiles, Engineering and Business Swedish School of Textiles, University of Borås, 501 90 Borås, Sweden; 3Laboratory of Instrumentation, University of Sciences and Technology Houari Boumediene, 16111 Algiers, Algeria; 4Assistive Technology Lab (NTA), Faculty of Electrical Engineering, Federal University of Uberlandia, Uberlandia 38408-100, Brazil; 5Laboratoire de Conception, d’Optimisation et de Modélisation des Systèmes (LCOMS), Université de Lorraine, 57000 Metz, France

**Keywords:** bladder monitoring, bladder urine volume, urinary incontinence, wearable devices, ultrasound, optical technology, electrical bioimpedance, urine leakage collection

## Abstract

Bladder monitoring, including urinary incontinence management and bladder urinary volume monitoring, is a vital part of urological care. Urinary incontinence is a common medical condition affecting the quality of life of more than 420 million people worldwide, and bladder urinary volume is an important indicator to evaluate the function and health of the bladder. Previous studies on non-invasive techniques for urinary incontinence management technology, bladder activity and bladder urine volume monitoring have been conducted. This scoping review outlines the prevalence of bladder monitoring with a focus on recent developments in smart incontinence care wearable devices and the latest technologies for non-invasive bladder urine volume monitoring using ultrasound, optical and electrical bioimpedance techniques. The results found are promising and their application will improve the well-being of the population suffering from neurogenic dysfunction of the bladder and the management of urinary incontinence. The latest research advances in bladder urinary volume monitoring and urinary incontinence management have significantly improved existing market products and solutions and will enable the development of more effective future solutions.

## 1. Introduction

The bladder is a muscular membrane organ that has the function of storing urine produced by the kidneys. The typical volume that the adult human bladder can support, before the urge to empty the bladder occurs, is estimated to be between 300 and 500 mL [1,2]. Bladder monitoring is the process of measuring and assessing the function and health of the bladder. It is an essential aspect of urological care, as it allows healthcare professionals to evaluate the bladder’s ability to store and empty urine and to detect abnormalities or problems that may be present [3].

There are various bladder monitoring methods, including invasive and non-invasive techniques for the diagnosis and management of a wide range of urological conditions, such as urinary incontinence, urinary tract infections, bladder cancer and voiding dysfunction. They can also be used in evaluating treatment efficacy and identifying potential alterations or adverse events during treatment [3,4,5,6,7].

Urinary incontinence (UI) management and bladder urinary volume (BUV) monitoring are two important aspects of urological care. UI, or the loss of control over urinary function, can have a significant impact on the quality of life (QoL) of those who suffer from it. Accurate and reliable diagnosis and UI management is essential for improving patient outcomes [8,9]. BUV monitoring, on the other hand, assesses bladder function and health by measuring the bladder’s capacity for storing and voiding urine, thereby enabling the identification of deviations or issues that may exist [3,10]. Both UI management and BUV monitoring could be performed using invasive and non-invasive techniques.

BUV monitoring has large clinical relevance, where an abnormal volume of urine within the bladder could indicate various health issues [4,5,6,7]. UI is one of the most common hygiene problems and affects millions of people around the world. Based on population studies from different countries, it has been reported that the prevalence of any type of UI is approximately 25% to 45%. The prevalence rates increase with age, and it has been found that the rates are even higher amongst nursing home patients. Furthermore, more than 40% of women over 70 years of age are affected by this inconvenience [11,12]. Different forms of UI exist, including, e.g., stress incontinence, which usually is the result of the weakening of the muscles used to prevent urination; urge incontinence, which is the result of overactivity of the detrusor muscles that control the bladder; and total incontinence, where the bladder is unable to store any urine at all [13,14,15]. Thus, it has been concluded that UI management is undoubtedly an important issue, where available and prescribed treatments for effective UI management depend on the type of incontinence, its severity, and the underlying cause [9].

Invasive techniques for UI management may include surgery, such as sling procedures, bladder augmentation, and artificial urinary sphincter. These methods involve the use of surgery to repair or replace the damaged structures that contribute to UI [16,17,18]. Invasive techniques for BUV monitoring are still widely used at hospitals—for instance, (1) urinary catheterization, which is the process of inserting a catheter through the urethra and into the bladder; (2) urodynamics, which measure the pressure and flow of urine during urination; and (3) cystoscopy, which uses a small camera to view the inside of the bladder and urethra. These methods involve the insertion of a device into the bladder or urethra to drain urine or measure its pressure and volume [19,20,21,22,23]. The use of invasive techniques offers highly accurate information; however, it is risky for the patient, who could experience discomfort, pain, perforation of the bladder, leaks, or a urinary tract infection [24,25,26].

Recent advancements in technology have led to the development of new and innovative non-invasive methods for both UI management and BUV monitoring, in addition to the several already well-known and used methods supporting patients suffering from UI. Examples include (1) pelvic floor muscle exercises (Kegel exercises), which can help to strengthen the muscles that control the flow of urine [27]; (2) bladder training, which can help to improve the ability to hold urine and avoid accidents [28]; (3) measurement of urine pH, urine flow rate, and post-void residual urine volume, which helps in the evaluation of bladder function [29,30], and (4) pharmaceuticals, which help to relax the bladder muscles and reduce symptoms of incontinence [31,32]. Wearable devices such as urinary collection devices and external catheters are also promising ways to improve UI management among females and older people. Some of these prototype products are designed to detect urine leakage and its amount in diapers or incontinence pads, while some only aim at discerning the number of urinations.

A study conducted by Root et al. [33] at a hospital emergency department has shown that the use of an External Urine Female Collection (EUFC) device is a good alternative to an Indwelling Urinary Catheter (IUC) and should be considered for the prevention of catheter-associated urinary tract infection [33]. Another effective solution to reduce diaper use among nursing home residents was Ultrasound-Assisted Prompted Voiding (UAPV) [34]. An ultrasound device was used to monitor the BUV, and patients were asked to urinate when the volume was greater than 75% of the individually prefixed bladder capacity. The results showed a significant reduction in absorption costs and an improvement in the caregivers’ working conditions, and the efficiency of UI management increased [35].

Technological advancements have facilitated the emergence of non-invasive techniques in monitoring bladder activity and collecting its associated information, by using various technologies including ultrasound, optical monitoring with Near-Infrared Spectroscopy (NIRS), and electrical bioimpedance [22,36,37]. These solutions tend to provide valuable information about bladder volume and function due to their ability to monitor the bladder over time, which offers a number of benefits, such as accurate information, ease-of-use, and improved patient comfort. Thereby, they can be used to improve the diagnosis and management of urological conditions. In this scoping review, the aim is to present the recent research advances in non-invasive technologies used on real human volunteers regarding bladder monitoring from an engineering perspective, with a focus on urinary incontinence management and bladder urinary volume monitoring. This will provide researchers, urologists, and healthcare professionals with a comprehensive understanding of the available non-invasive technologies, while highlighting their differences (material and technology) and defining their potential as an applicable solution for real life.

Hence, the scoping review focused on answering the following research questions: Which non-invasive technologies allow patients to perform BUV monitoring and UI management? What is the current SoA of BUV monitoring and UI management? What needs to be studied further in the context of BUV monitoring and UI management?

## 2. Methods

The methodological framework used in this work is a scoping review, which is described in detail by Tricco et al. [38]. The methodology was selected to gain a good overview of various recently developed technologies used for UI management and BUV monitoring, and to identify the advantages and disadvantages of these technologies.

### 2.1. Identification of Relevant Studies

To address the research questions, a methodical search of English-language articles published between 2015 and 2022 was performed in the PubMed, IEEE Xplore, ScienceDirect, Scopus, and Web of Science library databases. Initially, key terms such as bladder monitoring, urine incontinence, and wearable devices were used to identify relevant articles. The results of this initial search were used to optimize the search strategy and to refine the key terms used during the searches into more specific terms, including ultrasound, optical technology, electrical bioimpedance, urine leakage collection, and technologies that have undergone testing on human subjects.

The articles considered aimed to present information on (1) non-invasive technologies applied for BUV monitoring and UI management, (2) the basic characteristics and recent advancements of these technologies, and (3) information on other technologies widely utilized for BUV monitoring and UI management. To summarize, only articles that presented non-invasive technologies intended for use in human volunteers or patients were considered in the review. Studies involving animals, invasive technologies, phantom evaluations, and case reports were excluded.

### 2.2. Article Selection

The articles included in this scoping review were selected through a four-step process, which included identification, screening, eligibility, and the final selection of articles. In the identification step, the first (A.H.), second (S.D.), and third authors (C.A.) together reviewed the titles and abstracts of the identified articles. The authors (A.H.) and (S.D.) reviewed the full-text articles in the screening and eligibility step. All authors participated in the last step by ensuring that the articles included in this scoping review fulfilled only the inclusion criteria, and not the exclusion criteria. The literature search for this review was conducted from June 2021 to October 2022. Figure 1 presents the PRISMA flow diagram of the process followed for the selection of articles constituting this work.

### 2.3. Charting of Data

Initially, data were catalogued and sorted using Mendeley 2.6 and a spreadsheet. Data were summarized and entered in the spreadsheet, and organized by year of publication, aims of the article, technology, use of healthy volunteers or patients, and important results.

### 2.4. Summarizing and Reporting of Results

In contrast to a systematic review, a scoping review identifies a broad range of studies irrespective of their study design and quality, in order to present an overview of all the material selected for inclusion in the review [38]. Descriptions of the purpose and types of technology used for UI management and/or BUV monitoring were provided. The methodology used, technical details, population targeted, and the results obtained in each study were also described.

## 3. Results

This section is divided into two main parts: Section 3.1 presents technologies related to non-invasive BUV monitoring, and explains their modes of operation and the latest advances made in each of them, while Section 3.2 deals with solutions for UI management.

The database searches resulted in the identification of 207 articles, out of which 34 met the inclusion criteria for further analysis.

### 3.1. Bladder Urine Volume Measurement and Monitoring

The techniques employed for the non-invasive monitoring of BUV are diverse. The most common ones are ultrasound technology, optical technology, and electrical bioimpedance technology. This section presents the different main characteristics of each technology and provides examples of other technologies used for BUV monitoring.

#### 3.1.1. Ultrasound Technology

Ultrasound technology is an imaging technique that provides a wealth of information related to voiding dysfunction. The working principle of ultrasound is based on sending out and receiving a sound wave at a frequency that is beyond the range of human hearing. An ultrasound wave transmitter and receiver are attached to the skin above the bladder and used for scanning the bladder and for determining its shape. The ultrasound technology method of operation can be seen in Figure 2.

Ultrasound technology’s ability to visualize the bladder and prostate helps in the identification of several issues, such as intravesical prostate prominence or detrusor muscle diameter, and in revealing the presence of diverticula (small pockets that can develop in the bladder wall and stones that can cause blockages and pain) [39,40,41,42].

The first wearable ultrasound device for bladder monitoring was developed by Petrican et al. in 1998 [43]; since that time, the easy accessibility to commercial portable ultrasound devices, combined with the fact that the technology is non-invasive, safe, and painless, makes it one of the optimal BUV monitoring modalities [22,37,44].

Over the last few decades, advancements in technology have led to the improvement and optimization of ultrasound devices for BUV monitoring [3,37,45,46,47,48]. Van Leuteren et al. [45] presented another wearable ultrasound device called SENS-U, which is positioned on the lower abdomen using a skin-friendly adhesive. It is based on a combination of four ultrasound transducers within a field of view of 30°, which transmit ultrasound waves in the direction of the bladder perpendicularly to the abdominal wall. The sensor continuously estimates the BUV status and notifies the user to empty the bladder when the bladder is full. SENS-U’s clinical performance was evaluated using 30 children during urodynamic tests (which involved a diagnostic study of pressure in the bladder), where a set of tests consisting of measuring lower urinary tract function was performed. In the studied population, the device was able to detect a full bladder with a success rate of 90% and to notify the user almost 90% of the time. However, it was also reported that proper sensor positioning and childhood obesity are influencing factors that should be taken into account when using the device in daily clinical practice [45].

One of the most advanced improvements was also proposed by van Leuteren et al. [46], who developed a wearable and wireless ultrasound device named the URIKA bladder monitor (UBM). It was mainly dedicated to the detection of the total filling of the bladder in children that have dysfunctional voiding. For evaluating UBM, a study on 14 children with dysfunctional voiding, who were scheduled for clinical bladder training at a department of pediatric urology, was conducted. UBM estimates the anterior–posterior bladder dimension. When it exceeds a critical threshold, the patient is notified of a full bladder. It has been reported that UBM has an average accuracy rate of 85% in detecting bladder fullness. The detection rate among patients older than 10 years of age was 71%, while the detection rate among patients younger than 10 years of age was 100% [46].

A pilot study on 18 participants using the DFree device was conducted by Hofstetter et al. [47]. DFree is a small and portable device that uses ultrasound technology, and it has to be attached to the lower abdomen to measure BUV. The study aimed to investigate the impact of such ultrasound devices on the QoL and satisfaction of patients, by evaluating their usefulness, ease-of-use, and the degree of autonomy offered. The participants had various types of bladder dysfunction and used the DFree device for at least 12 h a day over a 3-month period. The conclusion of the study suggests that the device may be beneficial as a support for patients with bladder dysfunction; however, further technical development is needed to improve its reliability [47].

Table 1 presents a multidimensional comparison of the three commercially available portable ultrasound devices, UBM, SENS-U, and DFree, in terms of portability, their use for real-time analysis, participant groups used during testing, and their accuracy in detecting a full bladder.

Fournelle et al. [48] have introduced a novel low-cost, portable ultrasound device for research purposes, called MoUsE. It is used for long-term and automated BUV monitoring utilizing machine learning segmentation techniques. The device comprises 32 transmitters and receivers, and a 32-element phased array having a 3 MHz transducer. It is based on data digitization for signal reconstruction and subsequent image processing, where all reconstruction algorithms are executed on the GPU, enabling real-time reconstruction and imaging. To evaluate the accuracy of the trained algorithm and the system, MoUsE was used to acquire different bladder filling levels from four volunteers. The results indicate that the proposed approach demonstrates sufficient sensitivity for applications where a significant increase in BUV can occur. However, for applications requiring a precise quantitative assessment of BUV, further enhancement of the performance is necessary. Additionally, further investigations are required to enhance the performance of the system, with a particular focus on increasing the neural network size to enhance the system’s functionality on mobile devices with limited computing resources. Despite these limitations, the results obtained suggest that MoUsE could be a valuable tool for research and educational purposes in ultrasound imaging [48].

#### 3.1.2. Optical Technology

NIRS is an optical technology used for BUV monitoring applications. It is based on the measurement of light absorption, which is proportional to the fullness of the bladder. The system, which is usually placed on the skin above the midline of the lower abdomen, is constituted by a LED light source that has specific wavelengths for light emission and a photodetector that measures the absorption of the emitted light [49,50].

The technique allows for the detection of real-time changes in oxygenation in tissues via the measurement of chromophore changes [49], BUV, and other bladder parameters, such as lower urinary tract dysfunction, bladder outlet obstruction, and an overactive and/or underactive bladder, which can be measured and estimated using NIRS [37,50]. A schematic of the optical technology’s method of operation can be seen in Figure 3.

The NIRS technique has been studied by Macnab et al. since 2007, for several bladder functions, such as bladder outlet obstruction, overactive bladder, underactive bladder, or BUV [51].

A continuous wave NIRS instrument prototype for bladder monitoring has been designed, described, and tested in rural African clinics by Macnab et al. [52]. The design was optimized to enhance photon transmission and minimize interference from skin characteristics such as melanin pigmentation and hair. The prototype was equipped with three dual-wavelength LEDs, a silicon photodiode detector, and a long-pass optical filter to improve photon migration and skin contact. It was powered by either a rechargeable battery or AC power and communicated with a controlling computer via Bluetooth. The pilot testing was performed on 15 male volunteers, aged 21–87, who were recruited from a rural medical clinic. The results demonstrated the reproducibility of chromophore concentration data and showed that changes in chromophore concentration in the detrusor muscle during voiding could be monitored in pigmented subjects using NIRS. The successful monitoring was attributed to the device’s technical specifications, optimized application methodology, and a new protocol that improved device positioning and reduced signal contamination [52].

Fong et al. [53] investigated the use of NIRS in developing a wearable, BUV sensing device that provides alerts to patients with spinal cord injuries. The optical components used in their measurement system consisted of multiple high-power LEDs with a peak wavelength of 970 nm and a monolithic silicon photodiode. They performed an experiment that consist of a 1-min measurement on a healthy volunteer immediately before urination (i.e., when the bladder was full, to measure the light intensity through the diffuse reflectance data), and a 1-min measurement after urination to determine if any change in light intensity appeared. They observed that there was a noticeable drop in the light intensity detected by the photodiode between the full and empty bladder datasets obtained. The experimentation performed helped to demonstrate the feasibility of obtaining a reasonable signal using the setup with LEDs. This is an important step towards using the system in clinical trials on additional human volunteers [53].

Macnab et al. [54] also developed a system for wireless NIRS measurements of the bladder and brain simultaneously. The system consisted of two parallel dual-wavelength NIRS devices and proprietary software. The study monitored natural bladder filling and spontaneous voiding in two volunteer subjects using a 23-channel array over the frontal cortex and a 4-channel grid over the bladder. The results showed that simultaneous brain and bladder data were captured, indicating localized brain activity during bladder sensation and function. This study provides new physiological dimensions for evaluating bladder control and function [54].

#### 3.1.3. Electrical Bioimpedance Technology

Another promising technology is the use of electrical bioimpedance, where the BUV is estimated by measuring the impedance variation of the bladder during its activity. The technology is used to measure the electrical properties of biological tissue, and it is based on applying a small high-frequency electric current on the specific body segment, and measuring the resulting voltage generated from that segment, using conventional or dry electrodes [22,37,55]. Several studies have used electrical bioimpedance technology to obtain different information, such as the bladder state (full or not full) at any time, and for the monitoring of BUV [3,36,56,57,58,59,60,61,62,63,64,65,66]. A schematic of the electrical bioimpedance technology’s method of operation can be seen in Figure 4.

In an effort to develop a device that is appropriate for people with UI while in a seated posture, Sakai et al. [56] implemented an electrical bioimpedance-based prototype system to estimate BUV and conducted an experiment. The impedance value was measured using the SoC AD5933 (Analogue Device, Wilmington, MA, USA) at the frequency of 50 kHz, through two electrodes placed 5 cm from the centerline of the body, on a 24-year-old healthy male volunteer. Through experiments, they showed that there was a gradual decrease in the impedance value over time when the bladder was filled, even when body movement occurred. Thus, they could approximate the BUV from the impedance value obtained and compare it with the BUV obtained through measurement with conventional ultrasound device monitoring. They also observed a difference in the period during which urine storage could be tolerated and suggested to consider the tolerance parameter of urine storage in future studies [56].

A study conducted by Wang et al. [57] used electrical bioimpedance spectrum analysis to reveal the rhythmic neurogenic activity during natural bladder filling and detect urination desire. They placed two excitation electrodes and two measurement electrodes on the hypogastric surface of the body. To continuously measure changes in the BUV during different control stages of bladder filling, a 200 µA alternating current with 50 kHz frequency was required and provided by an electrical bioimpedance measuring device. The measurement also required wet Ag/AgCl electrodes with low polarization and an LCR meter to measure the voltage. The experiment was carried out on 12 healthy male volunteers, who had been asked not to drink tea, coffee, or soda prior to the experiment, and to empty their bladder and then immediately drink 100 mL of water as quickly as possible before starting the measurement. The results showed that, during natural bladder filling, there was a decrease in higher-frequency spectral power (0.15–0.4 Hz) and an increase in the low-to-high frequency ratio, whereas changes in the lower-frequency spectral power (0.04–0.15 Hz) were insignificant. Through their results, Wang et al. concluded that the method is useful in situations where evaluating the need to void and assessing neural regulation during bladder filling is of interest. Based on this study, it was found that there were significant changes in electrical bioimpedance spectroscopy during three stages: the first sensation, the urgency, and the discomfort stages [57].

Noguchi et al. [58] worked on a BUV measurement circuit with an electrical bioimpedance tetrapolar configuration, which can infer (1) change tendencies in impedance corresponding to BUV and (2) change tendencies in phase difference corresponding to the shape of the bladder. In this proposed circuit, four Ag/AgCl electrodes were placed to the left and right of the bladder, a sine wave current with a frequency of 50 kHz was injected, and a microcontroller (Arduino Due) was used to measure and calculate the voltages. Experiments were performed on a healthy 26-year-old male. Prior to the start of the experiment, the volunteer drank a total of 1.5 L of an isotonic beverage and urinated several times. In the 155-min-long experiment, only the phase difference was measured, but the authors state that it is possible to place electrodes around the abdomen and introduce a multi-channel phase difference measurement. Using such a setup, they expect to be able to extract detailed changes in the shape of the urinary bladder due to the use of an electrical impedance tomography measurement model and present these changes to the user [58].

Reichmuth et al. [59] developed a version of an electrical bioimpedance sensor system proposed in the future for measuring BUV during postsurgical surveillance. The battery-operated wearable and wireless system, which requires low power, uses four electrodes placed on the lower abdomen over the bladder. The measurement was performed on a healthy volunteer that drank a total of 1 L. The total duration of the experimental measurement exceeded 180 min. The results show that the device could be used for BUV measurement with only 80 µW power consumption at a 3 mHz sampling frequency. The low power consumption makes the device suitable as a long-term monitoring system [59].

Palla et al. [60] aimed to estimate the BUV using an electrical bioimpedance-based wearable system. The developed a device called Body Gate Way, which records the electrical bioimpedance signal through a disposable patch on the surface of the user’s skin at a sampling frequency of 32 Hz, and the signal from a 3D accelerometer at a sampling frequency of 50 Hz [60]. The sensitivity of the measurement was maximized by attaching medical electrodes to the lower part of the volunteer’s abdomen. The authors found that there was a visible trend in the electrical bioimpedance during the bladder filling process, although the presence of random noise, such as artifacts due to the volunteer’s movements, decreased the reliability of the measurement. To overcome this inconvenience, a Kalman filter was designed [61]. The filter is based on the constant velocity model, where the bladder volume ($V_{x}$) and the urinary flux that fills the bladder $(\dot{V_{x})}$, respectively, are its state variables. The results obtained have proven the validity and effectiveness of the proposed solution. This encourages the implementation and the testing of this solution in a real-world scenario [61].

A study on preferable electrode placement for accurate BUV measurement has been performed by Li et al. [62]. They realized a simulation study, based on a two-dimensional computational model, to determine the preferable tetrapolar electrode configuration locations for BUV measurement. After this, an experiment with eight young healthy volunteers, who were asked to drink a total of drink 1 L of water and lie on a bed, was performed in order to validate the simulation result and to investigate the correlation between the BUV and the measured electrical impedance values. The results obtained suggested that there was a strong negative correlation between the measured voltages and the BUV during bladder activity. They also noticed that both the simulation study and the study with humans suggested that the leftmost and rightmost points of the abdomen were the most preferable points to place the two electrodes that inject the AC current, and that it was preferable to place the two voltage sensing electrodes approximately 3 cm from the center of the abdomen [62].

In another work, Li et al. [63] proposed the use of electrical bioimpedance technology as an imaging technique. A system developed for BUV imaging, which is based on the Electrical Impedance Tomography (EIT) technique, was introduced. EIT is a specific application of electrical bioimpedance technology. Sixteen electrodes were used in the measurement. To validate the system, a study was conducted on six healthy volunteer subjects, and a parameter called the average conductivity index was derived from the EIT images. The results showed a high, positive, linear correlation between the average conductivity index parameter and the BUV in all subjects (correlation coefficient R = 0.98 ± 0.01). In their study conclusions, they state that EIT can be used for estimating BUV and that it has potential as a practical technique for assessing BUV [63].

Leonhäuser et al. [64] conducted a comparison study of the EIT technique with ultrasound technology for the measurement of BUV. For this, EIT measurement was performed using a commercial device (Goe MF II) that uses 16 electrodes placed at the level of the lower abdomen of the subject. Ultrasound technology was used to estimate the maximum BUV and residual urine using two methods, which were US-Ellipsoid/US-L × W × H and a bedside bladder scanner. Ten healthy volunteer subjects (5 women and 5 men aged 31 ± 5 years) participated in the comparison study. The results obtained revealed that there was a large offset between the measurements and actual BUV. In comparison to ultrasound technology, EIT shows considerable potential. However, the accuracy of EIT needs to be improved by minimizing the influence of movement artefacts and by optimizing the electrode placement [64].

In another study, Noyori et al. [65] were inspired by the EIT technique to estimate BUV by using eight-electrode electrical impedance measurements. They developed a device based on the HUZZAH32 board, and the system on chip AFE4300, which is an integrated analogue front-end that uses a tetrapolar I-V method for each electrical impedance measurement channel. The amplitude used for the AC current was set at 833 μA_p-p_ at a frequency of 50 kHz and a resolution of 17 mΩ. The proposed small-sized device has been used for continuous measurement in a feasibility test with a young healthy volunteer. The results obtained indicate that BUV can be estimated using the eight-electrode electrical bioimpedance measurement technique [65].

Body Impedance Analysis (BIA) is an application of electrical bioimpedance technology. Shin et al. [66] designed a comfortable waist-belt-type device working as a continuous BUV monitoring system based on BIA. The sensor was worn on the abdomen and connected to the body through Ag/AgCl electrodes. A Samsung Bio-Processor was used as the sensor circuit for impedance measurement. In their study, they injected a 50 kHz current to monitor the bladder, and used a 10 kHz current source for their algorithm, both frequencies at a current amplitude of 100 μA_p-p_. To minimize the inevitable movement artefacts, they suggested a motion artifact reduction algorithm that exploits multiple frequency sources. The experiments were performed on three healthy volunteers and the results indicate that there is a close relationship between BUV and impedance variation. This confirms the feasibility of their system for detecting enuretic events [66].

Table 2 provides an overview of the included studies on electrical bioimpedance technology for BUV monitoring, highlighting the main findings and setups of the studies.

#### 3.1.4. Other Technologies

Another interesting work providing information on recent progress regarding UI management is a study conducted by Rodas et al. [67], which estimated the emptying of the bladder in paraplegic and elderly people. To correctly determine when it is time for people suffering from spinal injuries to empty their bladder, they suggested a non-invasive method that consists of combining electrical bioimpedance, the hypogastric region temperature, and an artificial feedforward neural network. For electrical bioimpedance measurements, four Ag/AgCl electrodes were used, and a sine wave current with a frequency of 50 kHz and 1 mA_p-p_ was injected. The bladder region temperature was measured in real time using the chip MAX30205. Data obtained from more than 18 patients when emptying their bladder were then fed into a three-layer feedforward neural network with the BFGS Quasi-Newton algorithm. The results from patrons’ identification showed that there was a high correlation between electrical bioimpedance and the hypogastric region temperature when emptying the bladder. While the bladder is full, electrical bioimpedance is lower and the temperature is higher, and when the bladder is emptied, electrical bioimpedance is higher and the temperature is lower. The neural network had high accuracy of 99.80% and the reported mean square error was very low (1.08 × 10^12^) [67].

Kurihara et al. [68] developed a model for BUV measurement that requires no attachment of sensors to the skin. Instead, a hyperspectral camera (Resonon Inc: PikaXC2), which can take images at wavelengths ranging from 398.67 nm to 1016.78 nm, was used. The prediction was based on the idea of measuring the absorption spectrum of urine obtained immediately after urination. They performed a series of experiments on one healthy volunteer to evaluate the proposed method. They calculated its error rate based on the actual BUV and predicted urine volume using the method. The error rate of the proposed method was compared with the error rate for an ultrasound device attached to the skin above the bladder. The average error rate for the proposed method was 15.46%, whereas that of the ultrasound device was 23.42%. Hence, the proposed method was more accurate than the ultrasound device [68].

### 3.2. Urine Leakage Collection and Detection

The research on UI management aims to gather and analyze various technique-based solutions, in order to reduce the dependence on IUCs and traditional pads, by substituting them with external urinary catheters, while others aim to develop smart pads and underwear for urine leakage detection. Wearables, external catheters, and the most employed techniques for urine leakage detection are introduced in this section.

People suffering from UI use different types of urinary catheters, which are inserted in different ways to collect urine. However, patients favor external catheters due to their availability for home use and non-invasive characteristics. For instance, Beeson et al. [69] worked on an EUFC that consisted of an ultra-soft wicking fabric that absorbs and diverts urine away from the skin. They concluded that it was a good alternative to an IUC and that its use should be considered for the prevention of hospital-acquired conditions [69]. Sakamoto et al. [70] developed a self-powered wireless UI sensor system that determines the amount of urine in a diaper to a resolution of 100 cm^3^ in approximately 7 min. The system’s urine-activated battery is composed of two long, flexible electrodes. One electrode is based on activated carbon and the other one is based on aluminum. During a measurement, both electrodes are placed under a piece of absorbent material with a trench structure in a diaper [70].

In another project, Long et al. [71] introduced a solution to UI called TACT3. They developed and manufactured an underwear prototype capable of alerting the wearer to a pad leak before it reaches the outer clothing. It is a pair of washable fixation pants with sewn-in conductive threads that track the locations at which the pads leak most frequently. The conductive threads are connected to a removable signaling unit that vibrates three times when any part of the conductive threads becomes wet. In their experiment, 81 female participants, aged 67 years on average, were asked to wear the underwear prototype for a period of 2 weeks. More than 90% of the participants rated the overall impression of the prototype product as “good” or “OK”. The efficacy and acceptability of the evaluated underwear prototype were confirmed by the participants, who also reported a positive psychosocial impact [71].

As reported in Section 1, a work on implementing an EUFC at an emergency department was conducted by Root et al. [33]. They implemented a cylinder-shaped catheter covered by a soft wicking padded material that was meant to be placed externally over the female genitalia. When the patient voids, the urine is suctioned away from the patient to a wall canister. This keeps the patient’s perineum dry and free from urine-related moisture. Over a period of 3 months, 187 EUFCs were used on female patients. They noticed that the patients did not have any skin irritation during the whole period, which demonstrated that it was feasible to replace IUCs with EUFCs at the emergency department [33].

A low-cost wearable solution for detecting urine leakage and analyze urine biomarkers in normal daily life has been developed by Su et al. [72]. The system integrated an electrochemical biosensor in the diaper, which was able to detect moisture and chemical properties or biomarkers; a portable detection device; and a data processing application for a smartphone. A healthy volunteer wore the wearable solution to allow for real-time urine analysis. The results obtained showed high sensitivity, linearity, and selectivity for the detection and analysis of urine. A comparison made with the traditional instrument-based urine analysis in a hospital showed that the developed solution has the advantages of providing more convenient operation and flexible data access and that it is applicable in the field of urine leakage monitoring and urinary biomarker analysis [72]. The comparison made with the traditional instrument-based urine analysis in hospital, showed that the developed solution has the advantages of providing a more convenient operation and flexible data access and has shown applicability in the field of urine leakage monitoring and urinary biomarker analysis [72].

## 4. Discussion and Conclusions

This scoping review attempts to outline the recent progress in bladder monitoring solutions by highlighting current non-invasive technologies and the future directions of research on UI management and BUV monitoring. Different technologies based on ultrasound are already being used, and, more recently, electrical bioimpedance has been reported as a modern technology to use.

A considerable number of solutions with high potential for urinary management and analysis, mostly dedicated to females, older people with neurogenic bladders, and UI, have been developed. These solutions are based on active external collection devices, smart pads, or underwear equipped with UI sensor systems. They also attempt to offer the main alternatives to IUC, and this is due to their ability to detect urine leakage and their possibility to provide a more precise time estimation regarding the need to change the diaper after urination. Ideally, these solutions tend to be portable, self-powered, wireless, and comfortable to wear. Evaluation of these solutions in real-life conditions with more volunteers is required for validation, and additional development work is still needed to scale up the process for mass production.

In the estimation and monitoring of BUV, ultrasound devices can present challenges for the user in terms of their size and lack of portability, despite their widespread use. These limitations typically restrict measurements to a clinical or medical setting and may also contribute to inaccuracies in the measurements. Studies have reported a measurement error of up to 10% when comparing voided volumes measured by ultrasound devices with those recorded using frequency volume charting tools [73], which is a medical technique used to record the frequency and volume of a person’s bladder voids. However, these devices remain widely utilized among patients with UI due to recent advancements in miniaturization and portability, and their availability on the market.

Despite the promising potential of NIRS for BUV monitoring, there is a lack of quantitative evaluations conducted to date. One significant limitation of NIRS is the requirement of a large, permanently attached sensing device for continuous measurement, which can be inconvenient for patients. This limitation has likely contributed to the limited research interest in utilizing this technology.

The electrical bioimpedance technique has been proposed as a practical method for estimating BUV in individuals. However, the accuracy of these measurements can be affected by factors such as the skin area, skinfold thickness, electrical conductivity of urine, and movement artifacts. Despite these limitations, reliable and accurate results have been obtained through proper electrode placement and appropriate selection of the intensity and frequency of the injected current. Moreover, the advances in textile electrodes, miniaturized circuit designs, and ultra-low power systems, along with the integration of machine learning algorithms, have the potential to promote the development of wearable electrical bioimpedance-based devices for BUV monitoring and UI management applications in the future.

Other methods for determining BUV in individuals, such as utilizing specific cameras or measuring physical parameters such as temperature, have been proposed. However, the application of these techniques in human subjects has not been extensively studied, making the accuracy and reliability of these techniques for BUV monitoring uncertain. Techniques such as infrared thermography, 3D imaging, and microwave bladder state monitoring have also been examined as non-invasive options for BUV monitoring, but research investigating their accuracy in human subjects is limited, and additional studies are required to fully comprehend their potential in real-world scenarios.

The development, acceptance, and use of modern technologies in real-life conditions are influenced by various parameters, such as accuracy, cost, and ease-of-use. This also applies to BUV monitoring and UI management technologies. Ultrasound technology, for instance, has moderate to high accuracy levels, but it requires high-frequency sensors and ultrasound gel, which increases the cost and may decrease the ease-of-use. On the other hand, optical technologies, specifically near-infrared spectroscopy (NIRS) devices, are inexpensive and easy-to-use, but they are not as accurate as ultrasound-based devices. Electrical bioimpedance technology seems to offer a good middle ground as it aims to be low-cost, easy-to-use, and highly responsive in detecting changes in the human body. However, the accuracy of electrical bioimpedance technology varies with the devices used and populations studied.

Wearable technology, e.g., external catheters, is a promising solution for UI management since it is non-invasive, easy-to-use, and effective in managing various types of UI. These devices consist of a small catheter that is located outside of the body and connects to a collection bag, allowing urine to flow freely. They are comfortable to wear, discreet, and can increase patient autonomy. However, it is important to note that while wearable technology is an emerging field that is growing rapidly and the results obtained so far are promising, they have, to date, been evaluated mainly in experimental laboratory conditions and only in studies with a few healthy subjects. Therefore, future research, involving larger study populations of individuals suffering from UI, and real-life conditions, is needed to fully understand the potential of using wearable technology for BUV monitoring. Additionally, suggestions or recommendations for avoiding specific problems encountered when using wearable solutions could be provided to ensure the best outcomes when using them. Specific problems such as skin irritation and infection could be minimized by (1) regular cleaning and disinfection of the catheter and the surrounding area using appropriate skincare products, and (2) monitoring the skin for any signs of irritation or infection. Furthermore, using devices made of materials that are less allergenic can also help to reduce the risk of skin irritation and infection.

To conclude, the advancements in non-invasive technologies for BUV monitoring and UI management have shown potential in improving the QoL for patients suffering from bladder dysfunction. While some technologies, such as ultrasound and pads, have been developed and are currently available on the market, ongoing research is still necessary to discover new and improved solutions.

The utilization of non-invasive techniques provides a safe and effective alternative to invasive procedures and enables healthcare professionals to evaluate the bladder’s ability to store and empty urine, and to detect different abnormalities or problems that may be present. Therefore, further investigation in this field is essential to ensure that patients have access to the most optimal options for managing their bladder dysfunction.

The integration of machine learning techniques, a subfield of artificial intelligence, is a promising approach to improving the accuracy and precision of BUV monitoring systems. Furthermore, further research aimed at developing personalized treatment plans for individuals with UI, considering factors such as age, gender, and underlying conditions, is required. It is also expected that the e-health field will play an increasingly important role in UI management, due to its ability to facilitate the remote monitoring of patients’ conditions and provide a means for communication with healthcare professionals.

## Figures and Tables

**Figure 1 sensors-23-02758-f001:**
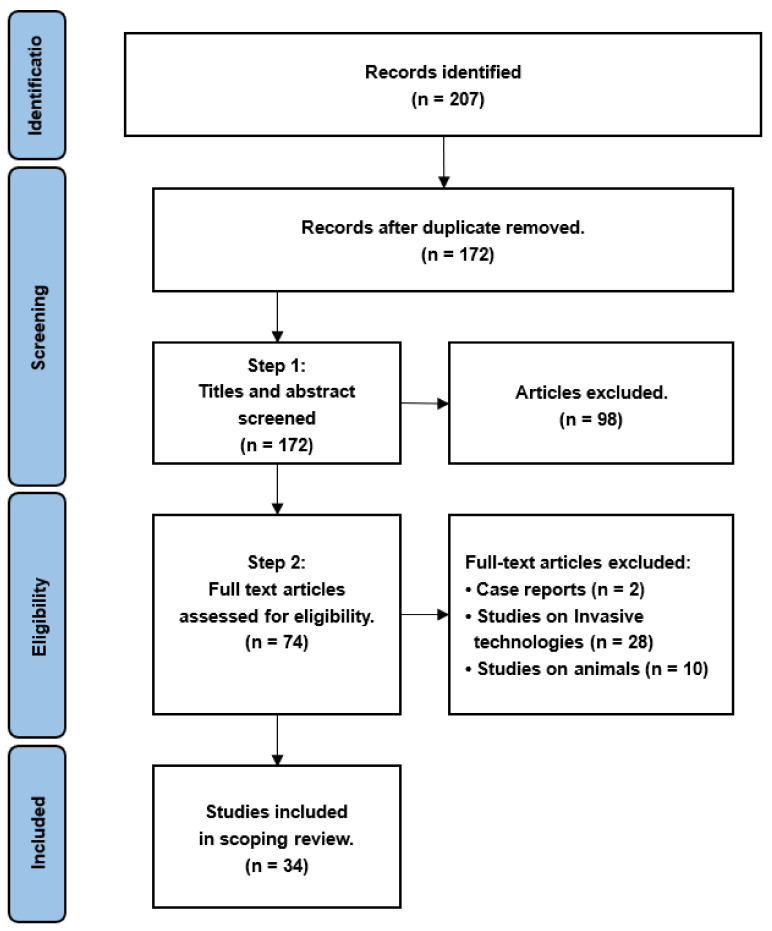
PRISMA flow diagram showing the process for selecting articles for inclusion in this scoping review.

**Figure 2 sensors-23-02758-f002:**
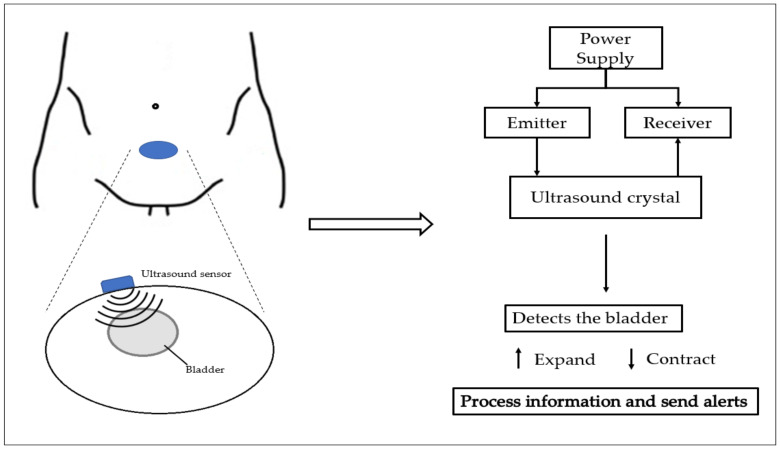
Schematic of ultrasound technology method of operation.

**Figure 3 sensors-23-02758-f003:**
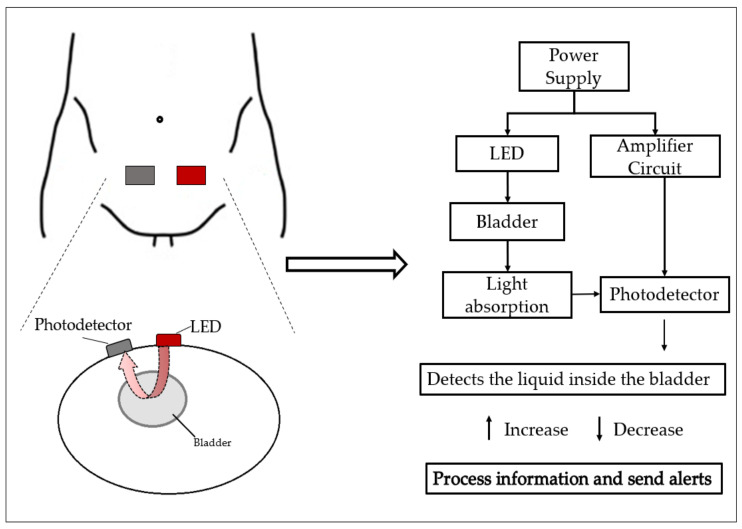
Schematic of optical technology method of operation.

**Figure 4 sensors-23-02758-f004:**
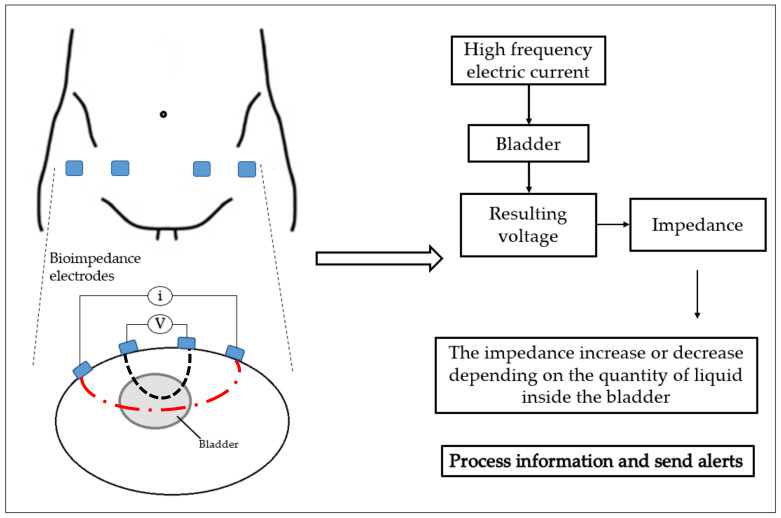
Schematic of bioimpedance technology method of operation.

**Table 1 sensors-23-02758-t001:** Comparison overview of the commercially available portable ultrasound devices UBM, SENS-U, and DFree.

Device	Wearable	Wireless	Real-Time Analysis	Tested in Children	Tested in Adults	Accuracy in Detecting Full Bladder
SENS-U	✓	✓	✓	✓	-	90%
UBM	✓	✓	-	✓	-	85%
DFree	✓	✓	✓	-	✓	-

**Table 2 sensors-23-02758-t002:** Overview of the included studies on electrical bioimpedance technology for BUV monitoring.

References	Year	Technical Characteristics (Current/Frequenc/N° Electrodes)	N° Healthy Volunteers	Results
[60]	2015	100 μA/50 kHz/4	1	There was a clear and detectable trend in the electrical bioimpedance during the bladder filling process, although the presence of random noise decreased the reliability of the measurement.
[57]	2016	200 μA/50 kHz/4	12	The method can be used to check the necessity to void since the higher-frequency spectral power (0.15–0.4 Hz) decreased (*p* = 0.05) and the low-to-high frequency ratio significantly increased (*p* = 0.001) during natural bladder filling.
[63]	2016	1 mA_p-p_/50 kHz/16	6	A high positive linear correlation between the average conductivity index parameter and the BUV in all subjects (correlation coefficient R = 0.98 ± 0.01), with the performance of the four-electrode method being much poorer (R = −0.27 ± 0.82).
[66]	2017	100 μA/10, 50 kHz/3	3	Results indicate that there is a close relationship between BUV and impedance variation. This confirms the feasibility of their system for detecting enuretic events.
[64]	2018	5 mA/50 kHz/16	10	The mean error of the ultrasound estimation methods (ellipsoid (37 ± 17%) and L × W × H (36 ± 15%) and EIT (32 ± 18%) showed no significant differences in estimating the maximum bladder capacity.
[58]	2018	500 μA/50 kHz/4	1	The circuit can infer the changes in BUV by measuring the electrical bioimpedance and phase difference of the urinary bladder. The rates of change for impedance and phase difference were different.
[62]	2019	1 mA_p-p_/10 kHz/4	8	There is a strong negative correlation between the measured voltages and BUV during bladder activity. The leftmost and rightmost points of the abdomen were preferable points to place the two electrodes that inject the AC current, and it was preferable to place the other two sensor electrodes around 3 cm from the center of the abdomen.
[56]	2019	-/50 kHz/-	1	The time for which urine storage could be tolerated was different in all the experiments. It was observed that the impedance value gradually decreased with passing time, even when body movement occurred.
[59]	2020	-/-/4	1	The device can be used for long-term monitoring since the results demonstrated the accuracy of the sensors and low power consumption of only 80 μW at 3 mHz.
[65]	2021	833 μA_p-p_/50 kHz/8	1	The portable device realized an SNR of 79.1 dB with a resolution of 0.017 Ω. It can estimate the BUV, although the estimation error was large when the voided volume was small.

## Data Availability

Not applicable.

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
