# Peer review of "State of the Art of Non-Invasive Technologies for Bladder Monitoring: A Scoping Review"

_sensors, 2023, doi:10.3390/s23052758_

Round 1

Reviewer 1 Report

Your work is focusing on new developments and tools in the follow up and diagnosis of Incontinence.  But actually you are treating a broad spectrum of urinary disorders. Not only incontinence but also voding disfunctions.

The last topic is only partially treated. 

The introduction need an urological clinical substantial revision.

This in mandatory From line 38 to 44 where urology clinic phisiology and patophisiology are vaguely mentioned

Actually, the two main objective of your study are diametrically opposing features of bladder conditions.

BUV measurement is fundamental in all very common obstructive conditions ( far well from urinary incontinence conditions spectrum) and togheter with urodynamics/PMR allow us to drive therapeutical choice and follow up after interventions and medical medications ( BPH, Chronic urinary obstruction, neurologic bladder, LUTS etc...)

In the other hand incontinence is varied clinical condition requiring monitoring of therapies effects (from drug administrations, to behavioral measures to surgical interventions).

Probably you have to add in introduction the existent invansive and non- invasive tools already existing and their innovation.

(eg: urodynamics probes, studies, software; recently developed app and techonologies for patient enrollment and control of symptoms and diary)

The work is focusing on engineering advancement and less on the clinical role of them. I think you have to modify your title focusing only on the probes (ultrasound/wereable etc.) and on the "Non-invasivity/wearebility".

Alternatively you should add paragraphs including other invasive and diagnostic tools used routinely in the study of incontinence and voiding dysfunctions.

Eg: No mention was done to electomiography or video-urodynamics ( historically used.

When you first refer to catheterization as the only method to obtain bladder volume you forget the routinary use of ultrasound in every health related ambient ( ER, outpatient clinics, at patient home, portable ultrasound and mobile app).

Moreover ultrasonography can give multiple informations relating to voiding dysfunction yet( Intravescical prostate prominence or detrusor muscle diameter, or revealing presence of diverticula and stones) in a non invasive way.

However is a very interesting work on these new avalaible technologies that could be usefull in a selected population ( eg: neurogenic bladder and paediatric population for the bladder outlet  conditions).

Finally replacing old fashioned measurement of incontinence (pads count pads weight, diary etc) with some wereable, connected devices has a big potential of application at today and in the next future.

Reviewer 2 Report

The manuscript provides a scoping review on the technologies and the current SoA of bladder urinary volume (BUV) monitoring and urinary incontinence (UI) management that concern both the urology community and the sensor research communities. Three main BUV monitoring technologies, namely ultrasonic systems, optical sensors and bio-impedance technologies, as well as smart wearable devices for UI management are addressed, identifying the advantages and disadvantages of each technology. The article is well-organized and well-written, citing carefully chosen literature publications, except that the conclusion and perspective are not strong enough. I therefore recommend the article for publication if the authors could make a minor revision by taking into account the following technical comments.

1.    In the abstract (line 21), “incontinence technology” could be changed to “incontinence managing technology”.

2.    In the introduction, the authors classified four types of UI and their causes (Lines 52-57), and state that “the effective management of UI depend on the type of incontinence, its severity, and the underlying cause”. The prostate hypertrophy is one of most common causes of UI in older men. The authors do not mention this category of UI and its management in this review. Would the authors care to comment?

3.    In the last paragraph of theUltrasound Technology” section, the authors presented two of the commercially available wearable incontinence ultrasound sensors URIKA and SENS-U. Other similar products exist, for example, the bladder sensor with bluetooth connectivity, named “DFree”, fabricated by Triple W, which won the “Best Award” for the Digital Health and Fitness Category at the International CES® 2019, mentioned in the reference [18]. As a review article, would it be possible to carry out a performance comparison between these sensors that could be helpful for urologists in their suggestion to patients, or is there any essential difference of them?

4.    In section “Bioimpedance Technology”, could the authors give a definition of bioimpedance to confirm that it is an electrical impedance, not other type of impedance, such the acoustical impedance that is used in ultrasonic technique?

5.    The authors devote a significant space to introduce the bioimpedance technology and the results obtained by numerous research teams. To facilitate readers' understanding, it would be appropriate to give a resume of these results, possibly in a table form, to compare their common points (50 kHz excitation for example), particularity, contribution, measurement method, and devices used, etc…

6.    It might be good to add a technical sketch (an original schematic diagram or authorized figure) for each of the three BUV monitoring methods discussed, which would greatly assist readers in understanding how these sensors work.

7.    In conclusion, the authors use the accuracy, cost, and ease of use as the criteria for future development of new technologies for BUV monitoring and UI management. For the BUV techniques reviewed, they conclude that: ultrasound technique shows good accuracy but requires high-frequency sensors and ultrasound gels, which impacts the cost and ease of use, while the optical technology is inexpensive and easy to use but not as accurate as ultrasound-based devices. The authors consider the bioimpedance technology as a good compromise with high sensitivity (does this equally imply high accuracy?) Does it mean that the bioimpedance technology will be the promising direction for the future? The solution for UI management is surly the wearable, external catheters technology. In future research and development, are there concrete suggestions or recommendations for avoiding some specific problems encountered, such as the skin irritation?

Reviewer 3 Report

Authors conduct a scoping review for bladder state monitoring.

My main concerns are:

- authors should state exacts searched keywords within the databases – without this information it is not eventually possible to replicate authors work, ideally please also provide number of papers in each database, and the date when you conducted the search,

- at several prats of the manuscript authors used references to 17-19 which are also reviews. As this is a review I would recommend replacing at several parts of the paper these references with the original articles, e.g. lines 152 and 216 – within the sentence authors mention “Several works” and then authors provide one reference. Further at line 140 there should be probably only one reference 17?

- paragraph 2.1. should be placed in my opinion in the last paragraph of introduction and then there should be several sentences describing what were steps authors did to answer those questions.

 - references should be added to paragraph line 58-63

Minor comments:

- why the searched time interval was 2015-2022 and not e.g. last 10 years?

- at line 248 you use “four … electrodes ” and in next paragraph at line 261 “4 electrodes” – please unite the style

- line 276 what is the Vx?

- line 411 “… future directions” this parts might be stressed a bit more – personally I would also include, e.g. microwave bladder state monitoring
